# Calibrating Reasoning in Language Models with Internal Consistency

**Zhihui Xie**[†§]   **Jizhou Guo**[†]   **Tong Yu**[‡]   **Shuai Li**[∗†]
[†]Shanghai Jiao Tong University    [‡]Adobe Research    [§]The University of Hong Kong
zhxieml@gmail.com   shuaili8@sjtu.edu.cn

## Abstract

Large language models (LLMs) have demonstrated impressive capabilities in various reasoning tasks, aided by techniques like chain-of-thought prompting that elicits verbalized reasoning. However, LLMs often generate text with obvious mistakes and contradictions, raising doubts about their ability to robustly process and utilize generated rationales. In this work, we investigate reasoning in LLMs through the lens of internal representations, focusing on how these representations are influenced by generated rationales. Our preliminary analysis reveals that while generated rationales improve answer accuracy, inconsistencies emerge between the model's internal representations in middle layers and those in final layers, potentially undermining the reliability of their reasoning processes. To address this, we propose *internal consistency* as a measure of the model's confidence by examining the agreement of latent predictions decoded from intermediate layers. Extensive empirical studies across different models and datasets demonstrate that internal consistency effectively distinguishes between correct and incorrect reasoning paths. Motivated by this, we propose a new approach to calibrate reasoning by up-weighting reasoning paths with high internal consistency, resulting in a significant boost in reasoning performance. Further analysis uncovers distinct patterns in attention and feed-forward modules across layers, providing insights into the emergence of internal inconsistency. In summary, our results demonstrate the potential of using internal representations for self-evaluation of LLMs.

## 1   Introduction

Large language models (LLMs) have demonstrated impressive capabilities in various reasoning tasks, aided by techniques like chain-of-thought (CoT) prompting that elicits verbalized reasoning (Wei et al., 2022; Merrill and Sabharwal, 2024). This approach directs the model to articulate step-by-step rationales before answering a question, simulating the reasoning process used by humans. With these verbalized rationales, it is expected that not only will the model's problem-solving capabilities be enhanced, but also the interpretability of its predictions will improve. Understanding how LLMs reason is essential for aligning them with human values (Bai et al., 2022; Li et al., 2024).

Despite the continued improvement in performance and the emergence of new capabilities, LLMs often generate text with obvious mistakes and contradictions, raising doubts about their ability to robustly process and utilize generated rationales (Ye and Durrett, 2022; Turpin et al., 2023). One notable failure mode is unfaithful reasoning, where LLMs provide rationales that contradict their final predictions (Lyu et al., 2023; Lanham et al., 2023). This makes it difficult to determine the trustworthiness of their predictions, highlighting the need for effective calibration methods to assess the reliability of rationales.

---

[∗]Corresponding author.

38th Conference on Neural Information Processing Systems (NeurIPS 2024).

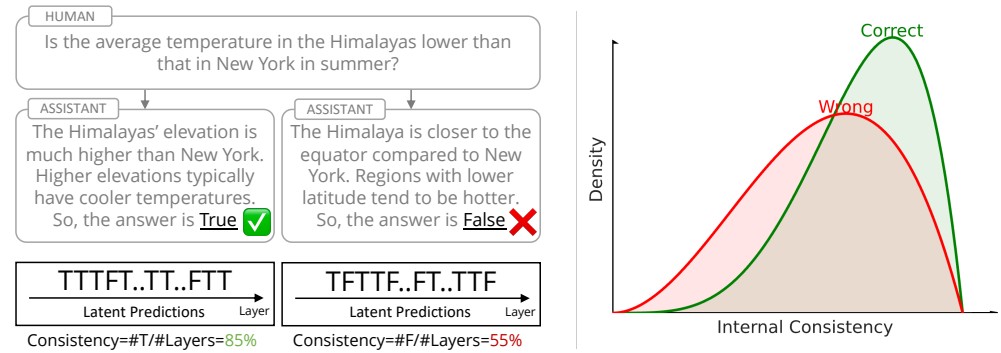

Figure 1: **An illustration of internal consistency.** Given a true-or-false question with the ground truth being true, we elicit latent predictions (i.e., predictions decoded from intermediate layers, represented by "T" for true and "F" for false) from the answer token of reasoning paths. By defining internal consistency as the agreement of latent predictions with the final stated one, we observe a high correlation between internal consistency and prediction accuracy, which aids in calibrating reasoning. Note that the figure on the right is synthesized for illustration purposes, but the distributions reflect actual observations as shown in Figure 3.

In this work, we present the first exploration of leveraging internal representations to calibrate reasoning in LLMs. Our study is based on the intuition that the inner workings of LLMs contain latent structures that can assess the reliability of reasoning paths. To investigate this, we begin by probing the model's intermediate activations for answer correctness in CoT reasoning. Our preliminary analysis reveals that CoT reasoning leads to inconsistencies between the model's internal representations in middle layers and those in final layers, potentially indicating a degree of uncertainty.

Based on this observation, we propose *internal consistency* (illustrated in Figure 1) as a measure of the model's confidence in the reasoning process. Inspired by previous works on eliciting latent predictions from intermediate layers (nostalgebraist, 2020; Belrose et al., 2023), we examine internal consistency by assessing the agreement of latent predictions decoded from intermediate layers. Unlike methods that require additional training and human annotations (Ye and Durrett, 2022; Khalifa et al., 2023), internal consistency provides a reliable and off-the-shelf self-evaluation for reasoning.

We conduct extensive empirical studies on various reasoning tasks, including reading comprehension, symbolic reasoning, and logical reasoning. Our analysis across different models and datasets demonstrates that internal consistency effectively distinguishes between correct and incorrect reasoning paths. Moreover, by up-weighting reasoning paths with high internal consistency, we achieve a significant improvement in reasoning performance. These results highlight the potential of using internal representations for the self-evaluation of LLMs.

This work makes three main contributions. 1) We identify the emergence of inconsistencies between intermediate and final layer representations in LLM reasoning, highlighting a potential issue where CoT reasoning leads to higher uncertainty. 2) We propose internal consistency as a novel measure to evaluate the model's confidence and calibrate reasoning by up-weighting paths with high internal consistency. 3) We provide insights into the cause of internal inconsistency in reasoning by analyzing Transformer components (i.e., attention and feed-forward networks) across layers. We believe our results show promise in leveraging internal representations to enhance reasoning in LLMs.

## 2 Preliminaries

In this section, we present background information, notations, and preliminary analysis to provide context for our study.

### 2.1 Transformer Architecture

Our analysis focuses on the prevalent decoder-only Transformer architecture (Vaswani et al., 2017; Radford et al., 2018). To set notation and context, we briefly describe the key components as follows.

Given a sequence $\mathbf{x} = (x_1, \ldots, x_n)$ of input tokens, the inference process of the Transformer begins by projecting these tokens into a sequence of representations $\mathbf{h}_1^0, \ldots, \mathbf{h}_n^0 \in \mathbb{R}^d$. This representation is then updated by a series of $L$ residual blocks (Elhage et al., 2021), each consisting of a multi-head self-attention (MHSA) layer followed by a feed-forward network (FFN) layer[2]. In each block $\ell \in [0, L-1]$, the representation of each token $i$ is updated as follows:

$$\mathbf{h}_i^{\ell+1} = \mathbf{h}_i^\ell + \text{FFN}^\ell(\mathbf{h}_i^\ell + \text{MHSA}^\ell(\mathbf{h}_i^\ell)).$$

Finally, the next token distribution is produced by applying an unembedding on $\mathbf{h}_i^L$:

$$p(x_{n+1} \mid \mathbf{x}) = \text{Softmax}(\text{Unembed}(\mathbf{h}_i^L))_{x_{n+1}}.$$

As the dimension of $\mathbf{h}_i$ remains constant across layers, we can view it as a dynamic distribution processed by the model (Geva et al., 2022), providing insight into how LLMs reason internally.

## 2.2 Preliminary Analysis of Internal Representations in CoT Reasoning

Building upon works that explore the inner workings of LLMs (Burns et al., 2023; Ferrando et al., 2024), our study starts with exploring the encoded information in internal representations during reasoning. In the context of CoT prompting—a prevalent technique for eliciting reasoning capabilities of LLMs—we analyze these representations along two dimensions: *horizontally* across reasoning steps, and *vertically* across network layers.

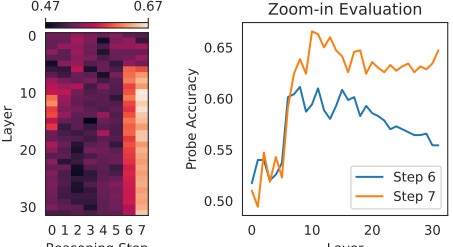

We conduct preliminary experiments using the Llama-2-7B model (Touvron et al., 2023) on the PrOntoQA dataset (Saparov and He, 2023), which challenges the model to determine whether a proposition is true based on a series of assumptions. We apply 4-shot CoT prompting and greedy decoding to generate reasoning paths and predictions. To discern encoded information crucial for reasoning, we utilize linear probing (Alain and Bengio, 2016; Belinkov, 2022) to train classifiers on the activations with respect to the ground truth labels, aiming to differentiate between true and false propositions. Reasoning paths are segmented into individual steps using the NLTK punkt sentence tokenizer (Bird et al., 2009), with a dedicated probe classifier for each layer at the last token of every reasoning step. Implementation details are provided in Appendix B.1.

Figure 2: **CoT reasoning improves answer accuracy but exacerbates the inconsistency between hidden and stated reasoning.** Left: Heatmap of linear probe accuracies at all reasoning steps and intermediate layers. Right: A zoom-in on the results of the last two steps.

Figure 2 presents an evaluation of the probe accuracy on the validation set, including a detailed zoom-in on the probe accuracy at the last two reasoning steps, which reveals two distinct patterns:

1. **Improved accuracy through verbalized reasoning:** Probe accuracy increases monotonically as the model processes and verbalizes reasoning paths, indicating that LLMs extract and utilize task-solving information from their generated rationales.

2. **Emergent inconsistency across layers:** A closer examination at the last two reasoning steps reveals a notable inconsistency in probe accuracy between middle and later layers. This inconsistency suggests that while middle layers capture essential reasoning information, it may not be fully utilized or maintained by the later layers, potentially impacting the model's overall reasoning performance.

This analysis provides an initial understanding of how LLMs internally process and handle information during CoT reasoning. While LLMs effectively gather information from verbalized reasoning paths, there is a noticeable tendency for these models to underutilize the processed information at intermediate layers. This motivates us to further investigate into the internal representations of LLMs and explore methods to better harness them to enhance reasoning capabilities.

---

[2]Modern Transformer models apply normalization (Ba et al., 2016; Zhang and Sennrich, 2019) before or after each layer. We omit the notation here for simplicity.

## 2.3 Calibration

Calibration refers to the alignment between a model's predicted probability estimates and their actual prediction correctness (Guo et al., 2017; Geng et al., 2023). Calibration is crucial for assessing the reliability and trustworthiness of LLMs, and can also help improve performance (Wang et al., 2022).

More formally, we want to find a measure Score such that for any two prompt-answer-prediction triplets $(\mathbf{x}_i, \mathbf{y}_i, \hat{\mathbf{y}}_i)$ and $(\mathbf{x}_j, \mathbf{y}_j, \hat{\mathbf{y}}_j)$, the following holds:

$$\text{Score}\,(\mathbf{x}_i, \hat{\mathbf{y}}_i) \leq \text{Score}\,(\mathbf{x}_j, \hat{\mathbf{y}}_j) \iff p\,(\hat{\mathbf{y}}_i = \mathbf{y}_i \mid \mathbf{x}_i) \leq p\,(\hat{\mathbf{y}}_j = \mathbf{y}_j \mid \mathbf{x}_j)\,.$$

In this work, we reveal the potential of using internal representations for calibration and introduce a new measure based on internal consistency, as described in Section 3.2. The new measure does not require additional training and is more accurate than logit-based methods.

# 3 Internal Consistency as a Lens to Calibrate Reasoning

In Section 2.2, we uncover patterns of internal inconsistency in reasoning with linear probes. While probe accuracy indicates the "extractability" of task-solving information, it does not necessarily show how LLMs process internal representations for generation (Alain and Bengio, 2016; Belinkov, 2022). Therefore, we introduce a new method to measure internal consistency in this section.

## 3.1 Interpreting Internal Representations

The internal reasoning process of a Transformer towards its final prediction can be comprehended through iterative inference (Jastrzębski et al., 2017; Geva et al., 2022). A direct method to inspect this internal process during generation is to early exit from intermediate layers. Specifically, instead of applying unembedding on $\mathbf{h}_i^L$, we can obtain the token distribution over the vocabulary from any intermediate layer using the logit lens (nostalgebraist, 2020):

$$p^\ell(x_{n+1} \mid \mathbf{x}) = \text{Softmax}(\text{Unembed}(\mathbf{h}_i^\ell))_{x_{n+1}}.$$

Building on this interpretation, we can decode *latent predictions* from any layer $\ell$:

$$\hat{x}^\ell_{n+1} = \arg \max_x p^\ell(x \mid \mathbf{x}). \tag{1}$$

While Equation 1 provides a measurement of latent predictions, we find that the decoded distributions are often miscalibrated, biasing towards specific answers (Zhao et al., 2021; Belrose et al., 2023). For instance, in the PrOntoQA task with Llama-2-7B, the penultimate layer consistently assigns over 90% probability to the answer "True," regardless of the context. To address this, we balance the latent predictions across the possible answers for each layer separately. See Appendix B.4 for more details.

## 3.2 Calibration Measurement with Internal Consistency

Built on our findings in Section 2.2, we investigate consistency of internal representations as an indication of uncertainty. Intuitively speaking, if the model's internal representations are highly inconsistent with those of later layers, it may not faithfully say as it thinks. Based on these intuitions, we propose a simple metric called *internal consistency* to measure the agreement of latent predictions.

**Internal Consistency** During inference, we collect latent predictions at the answer tokens and measure how often latent predictions match the final prediction:

$$\text{InternalConsistency}(\mathbf{x}, \hat{\mathbf{y}}) = \frac{1}{L-1} \sum_{\ell=1}^{L-1} \mathbb{1}\{\hat{\mathbf{y}}^\ell = \hat{\mathbf{y}}^L\}. \tag{2}$$

This measure offers a straightforward method to gauge the level of internal consistency, serving as a useful indicator of the output's correctness (i.e., calibration), without requiring additional training.

**Calibrating Reasoning with Internal Consistency**   As LLMs often require long reasoning paths to resolve complex tasks (Wei et al., 2022), a process that introduces cumulative errors and uncertainty, we integrate this consistency measure into the generation process to enhance reasoning calibration. Specifically, we propose a new decoding method that assigns weights to each sampled reasoning path based on their calculated consistency score at the final answer token. Paths that exhibit higher internal consistency—indicating robust alignment between intermediate and final predictions—are assigned greater weight in determining the final model output. This weighted strategy aims to prioritize reasoning paths that not only maintain self-consistency but are also more likely to converge on the correct answer, thus potentially enhancing the model's accuracy in complex reasoning tasks.

## 4   Experiments

To study internal consistency, we conduct extensive experiments to address the following questions: 1) To what degree does internal consistency correlates with reasoning performance? 2) Can we leverage internal consistency to boost reasoning? 3) How do different components of the Transformer architecture contribute to the emergence of internal inconsistency?

### 4.1   Setup

We evaluate internal consistency across a spectrum of models and tasks. See Appendix B.2 for further details of the experimental setup.

**Models**   Our experiments are conducted using two prominent series of open-source Transformer-based models: the Llama-2 series (Touvron et al., 2023) and the Mistral series (Jiang et al., 2023). Specifically, we evaluate both the 7B and 13B configurations of the Llama-2 series to understand how model scale impacts internal consistency. Additionally, for the Mistral series, we explore the 7B and the $8 \times 7B$ versions, the latter of which incorporates mixture of experts layers (Jiang et al., 2024). These models were chosen due to their extensive use and distinct architectural features, enabling a comprehensive evaluation of internal consistency.

**Datasets**   We evaluate various datasets that span reading comprehension, symbolic reasoning, and logical reasoning tasks. We choose these datasets because they involve explicit reasoning processes with unambiguous single-token answers (i.e., "True" and "False"), which facilitates our analysis. To standardize the evaluation process, each dataset is transformed into a true-or-false QA format. For a detailed description of the datasets and examples, please refer to Appendix B.2.

1. BoolQ (Clark et al., 2019): A reading comprehension dataset where each instance involves a yes/no question grounded in a related passage.
2. CoinFlip (Wei et al., 2022): A dataset that challenges the model's symbolic reasoning abilities by presenting a task where the model must determine the outcome of a coin (heads or tails) after a series of flips.
3. PrOntoQA (Saparov and He, 2023): A dataset designed for logical reasoning. Each question requires a 3-hop deduction to determine the truth value of a proposition based on a set of assumptions with fictional concepts.
4. ProofWriter (Tafjord et al., 2020): A logical reasoning dataset that, in contrast to PrOntoQA, uses real concepts for all assumptions. Each question requires 3 hops of reasoning.

We balance the labels and each dataset has at least 500 samples for evaluation. Our method requires no training procedure. To evaluate reasoning performance, we use calibrated accuracy following Zhao et al. (2021); Burns et al. (2023), balancing predictions to be 50/50 across the two labels.

### 4.2   Internal Consistency is a Good Calibration Measure

Our analysis begins by examining patterns of internal consistency across different layers of LLMs. As illustrated in Figure 6 (Appendix C), latent predictions exhibit variable convergence during inference; notably, there are significant increases in consistency in the middle and final layers, while the early layers are characterized by considerable noise. Additionally, these patterns vary across models and

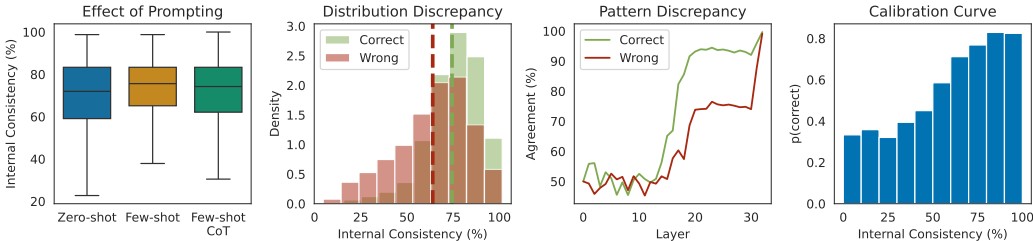

Figure 3: **Internal consistency is a reliable measure of prediction confidence in CoT reasoning.** From left to right: 1) the effect of different prompting techniques on the model's internal consistency; 2) the distribution discrepancy of internal consistency between correct and incorrect model predictions; 3) pattern variations in agreement values (representing the ratio of data instances where the latent predictions match the final predictions) across layers; and 4) a calibration plot with bins according to the model's internal consistency on the x-axis and the accuracy within each bin on the y-axis. Results are averaged over all models and datasets. See Appendix C for the full results.

datasets. Smaller models, such as Llama-2-7B, show high variability in latent predictions across layers, indicating less stability in their reasoning processes. In contrast, larger models like Mixtral 8×7B demonstrate more robust internal reasoning, as evidenced by more reliable convergence of latent predictions. For BoolQ, the inconsistency is less pronounced compared to other datasets that require more complex symbolic and logical reasoning.

Furthermore, the correlation between internal consistency and model accuracy is particularly evident when distinguishing between correct and incorrect answers in CoT reasoning. As depicted in Figure 3, the impact of prompting techniques on internal consistency is significant, highlighting the emergence of inconsistency brought by CoT prompting as discussed in Section 2.2. While few-shot demonstrations generally enhance consistency, CoT prompting decreases it. A closer examination of CoT reasoning reveals that reasoning paths leading to incorrect answers typically exhibit lower internal consistency. These results underscore the potential of internal consistency not only as a diagnostic tool but also as a reliable measure for calibrating reasoning in LLMs.

## 4.3 Enhancing Reasoning with Internal Consistency

Our analysis reveals a strong correlation between internal consistency and prediction accuracy, suggesting its potential as a mechanism for improving reasoning performance. As demonstrated in Section 4.2, CoT prompting, while effective, tends to decrease internal consistency. This decrease is possibly attributable to the generated reasoning paths that incorporate incorrect rationales, which confuse the model to make inconsistent predictions across layers. To investigate the generality of this phenomenon and its potential solutions, we also least-to-most (L2M) prompting (Zhou et al., 2022), which decomposes complex problems into simpler sub-problems for reasoning. For a detailed description of our least-to-most decomposition approach, please refer to Appendix B.3.

To incorporate internal consistency, we build on the self-consistency (SC) approach (Wang et al., 2022), which utilizes an ensemble of multiple sampled reasoning paths to increase accuracy. Our proposed method, which we term SC+IC, integrates internal consistency to weight these reasoning paths accordingly. Specifically, if a reasoning path demonstrates high internal consistency in its answer prediction, the probability of accepting its corresponding answer as the final answer increases. We accumulate the sum of internal consistency scores across all reasoning paths for each answer and choose the answer with the highest sum. For baselines, we also compare with greedy decoding Greedy and a logit-based approach SC+$\Delta$ (Wang and Zhou, 2024) that selects the final answer based on a confidence measure $\Delta_k = p\left(\hat{\mathbf{y}}^1 \mid \mathbf{x}\right) - p\left(\hat{\mathbf{y}}^2 \mid \mathbf{x}\right)$, where $\hat{\mathbf{y}}^1$ and $\hat{\mathbf{y}}^2$ represent the top two tokens for answer prediction of the $k$-th path.

Table 1 summarizes the results, which demonstrate that SC+IC consistently outperforms the others across different models and tasks, with improvements ranging between 1.8% to 4.9% across models. Figure 4 plots the calibrated accuracy with respect to varying numbers of sampled paths. We observe clear improvements of SC+IC over the baselines, highlighting the effectiveness of leveraging internal

Table 1: **Internal consistency improves reasoning performance across diverse tasks.** We report calibrated accuracy averaged across runs of 10 different random seeds. For self-consistency (SC) and its variants, results are obtained with 40 sampled reasoning paths.

| | BoolQ | | CoinFlip | | PrOntoQA | | ProofWriter | | Mean | |
|---|---|---|---|---|---|---|---|---|---|---|
| | CoT | L2M | CoT | L2M | CoT | L2M | CoT | L2M | CoT | L2M |
| **LLAMA-2-7B** | | | | | | | | | | |
| Greedy | 67.1 | 61.4 | 71.2 | 81.9 | **51.7** | 51.1 | 52.4 | 60.7 | 60.6 | 63.8 |
| SC | 70.0 | 73.2 | 76.0 | 89.2 | 49.0 | 50.0 | 52.2 | 66.8 | 61.8 | 69.8 |
| SC+$\Delta$ | 69.9 | 73.0 | 76.2 | 90.1 | 50.2 | 50.1 | 52.4 | 68.2 | 62.2 | 70.3 |
| SC+IC | 70.7 | 73.4 | 77.6 | 90.2 | 49.0 | 51.2 | 52.2 | 69.0 | 62.4 | 71.0 |
| SC+IC (tune) | **71.5** | **73.5** | 78.6 | **93.9** | 50.8 | **55.7** | 54.1 | 70.7 | **63.8** | **73.4** |
| SC+IC (transfer) | **71.5** | **73.5** | **78.7** | **93.9** | 50.8 | **55.7** | 53.9 | 70.7 | **63.8** | **73.4** |
| **LLAMA-2-13B** | | | | | | | | | | |
| Greedy | 78.5 | 71.3 | 78.9 | 90.5 | 53.6 | 54.0 | 60.8 | 78.1 | 67.9 | 73.5 |
| SC | 80.8 | 78.3 | 81.0 | 94.2 | 52.6 | 55.4 | 61.0 | 88.6 | 68.9 | 79.1 |
| SC+$\Delta$ | 81.2 | 78.9 | 81.8 | 94.2 | 55.0 | **57.6** | 61.6 | 87.9 | 69.9 | 79.6 |
| SC+IC | **81.4** | 78.7 | 84.0 | 93.2 | **55.4** | 57.0 | 61.8 | 88.6 | 70.6 | 79.4 |
| SC+IC (tune) | **81.4** | **80.4** | 86.2 | **96.5** | 54.5 | 56.6 | **62.9** | 89.6 | **71.2** | **80.8** |
| SC+IC (transfer) | **81.4** | 80.2 | **86.4** | 96.3 | 54.4 | 56.2 | 62.8 | **89.7** | **71.2** | 80.6 |
| **MISTRAL-7B** | | | | | | | | | | |
| Greedy | 74.0 | 72.7 | 89.8 | 99.1 | 55.3 | 53.4 | 65.6 | 73.3 | 71.2 | 74.6 |
| SC | 76.8 | 78.9 | 95.2 | 99.4 | 58.0 | 51.4 | 68.6 | 84.0 | 74.6 | 78.4 |
| SC+$\Delta$ | 77.9 | **79.5** | 95.2 | 99.4 | 59.6 | 52.7 | 66.8 | 85.3 | 74.9 | 79.2 |
| SC+IC | **78.1** | 78.9 | 96.4 | 99.4 | 59.8 | 53.2 | 70.0 | 85.0 | 76.1 | 79.1 |
| SC+IC (tune) | 77.8 | **79.5** | 97.7 | **99.9** | 60.4 | **56.6** | **71.2** | 88.6 | **76.8** | **81.2** |
| SC+IC (transfer) | 77.8 | 79.4 | **97.7** | **99.9** | 60.5 | 56.5 | 71.1 | **88.7** | **76.8** | 81.1 |
| **MIXTRAL-8×7B** | | | | | | | | | | |
| Greedy | 78.8 | 78.2 | 98.8 | 99.4 | 57.4 | 57.9 | 72.3 | 79.4 | 76.8 | 78.7 |
| SC | 81.6 | 85.3 | 99.4 | **100.0** | 61.6 | 56.8 | 75.6 | 90.2 | 79.6 | 83.1 |
| SC+$\Delta$ | **82.6** | 85.6 | 99.4 | **100.0** | 63.4 | 57.9 | 75.0 | 90.6 | 80.1 | 83.6 |
| SC+IC | 81.7 | 85.6 | 99.4 | **100.0** | 63.4 | 56.4 | 78.0 | 90.6 | 80.6 | 83.2 |
| SC+IC (tune) | 81.8 | **86.1** | 99.9 | **100.0** | **63.8** | 59.3 | 78.8 | **92.1** | **81.1** | **84.4** |
| SC+IC (transfer) | 81.8 | **86.1** | **100.0** | **100.0** | **63.8** | 59.5 | 78.9 | **92.1** | **81.1** | **84.4** |

consistency to enhance reasoning. This improvement is particularly notable in tasks involving symbolic and logical reasoning, which depend heavily on the correctness of reasoning paths. In addition, we observe that internal consistency effectively distinguish between correct and incorrect paths (e.g., in the PrOntoQA examples in Table 2, internal consistency helps filter out flawed rationales), validating our hypothesis on why internal consistency can enhance CoT reasoning.

## 4.4 Layer-weighted Aggregation Finds Transferable Patterns

While results of SC+IC have demonstrated the effectiveness of leveraging latent predictions for reasoning, the relative importance of different layers remains unexplored. Recent studies have shown that certain intermediate layers specialize in specific types of reasoning (Yang et al., 2024). These observations suggest that treating all layers equally when computing internal consistency might be suboptimal. Motivated by these insights, we introduce an aggregation strategy for internal consistency with two variants: SC+IC (tune) and SC+IC (transfer). Specifically, instead of assigning equal weights to each layer in Equation 2, we consider a learned weight vector $\mathbf{w} \in \mathbb{R}^L$ (where $L$ is the number of layers) for aggregation. For SC+IC (tune), we optimize layer weights using 500 held-out samples per dataset. The weights are learned using Adam optimizer with a learning rate of 0.01 for 1,000 iterations. SC+IC (transfer) evaluates cross-task generalization by applying weights tuned on PrOntoQA to other datasets.

The results are presented in Table 1 and Figure 4. Despite introducing only $L$ parameters, this weighted approach yields substantial improvements. Notably, the strong performance of SC+IC

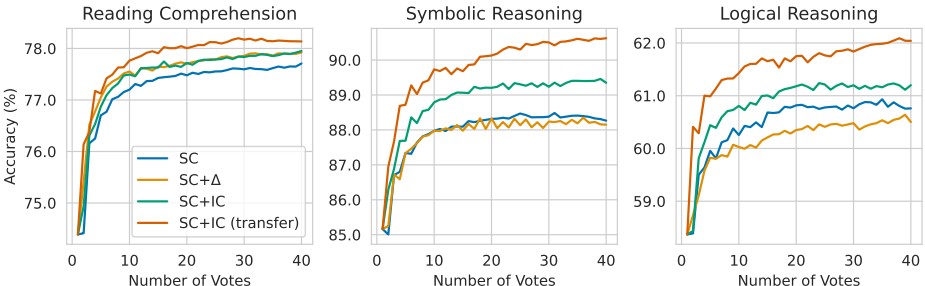

Figure 4: **Internal consistency brings larger gains for complex reasoning tasks.** The figure shows calibrated accuracy as a function of the number of votes for three types of tasks: reading comprehension (BoolQ), symbolic reasoning (CoinFlip), and logical reasoning (PrOntoQA and ProofWriter). Results are averaged over all models and random seeds.

(transfer) across diverse tasks indicates that optimal layer importance patterns generalize well, demonstrating the broad applicability of internal consistency for reasoning tasks.

## 4.5   Investigating the Role of Transformer Components in Internal Inconsistency

Having observed that CoT prompting exaggerates internal inconsistency and that reasoning performance can be improved by re-weighting reasoning paths based on internal consistency, we now aim to understand why internal inconsistency emerges from CoT reasoning. Specifically, we explore how different network components contribute to the final predictions in CoT reasoning and, consequently, the calibration effect of internal consistency. One hypothesis is that this effect arises from better elicitation of information processed in the middle layers, which tends to be underutilized in later layers. To investigate this, we conduct a detailed analysis focusing on the roles of self-attention and feed-forward network (FFN) layers.

**Self-attention Layers**   We analyze the average attention weights across all attention heads at each layer, focusing on what the answer tokens attend to. Consistent with the CoT process, we segment the processed tokens into three parts: the context, the query, and the generated rationale. Further details on the methodology and calculations can be found in Appendix B.5.

**FFN Layers**   For FFN layers, we use a probe vector trained on the model's last hidden state of the answer token with respect to the model's output. We compute cosine similarities between this probe vector and the *value vectors* (Geva et al., 2022; Lee et al., 2024), each corresponding to a column

Table 2: **Examples where internal consistency helps select correct reasoning paths for Mistral-7B on PrOntoQA.** We show the paths with the highest and lowest internal consistency. The question has been reformatted for better presentation. Incorrect steps are colored in gray, whereas wrong predictions are highlighted in red.

---

**Question**: Zumpuses are not transparent. Each zumpus is a wumpus. Wumpuses are red. Each wumpus is a jompus. Impuses are amenable. Every jompus is not spicy. Jompuses are tumpuses. Every tumpus is floral. Tumpuses are yumpuses. Every yumpus is not amenable. Every yumpus is a vumpus. Each vumpus is metallic. Every vumpus is a numpus. Numpuses are temperate. Numpuses are dumpuses. Dumpuses are not small. Dumpuses are rompuses. Polly is a jompus. True or false: Polly is amenable.

---

**Sampled Paths**:
**P1 (IC=0.875)**: Polly is a jompus. Jompuses are tumpuses. Polly is a tumpus. Every tumpus is floral. Tumpuses are yumpuses. Polly is a yumpus. Every yumpus is not amenable. Polly is not amenable. False

**P2 (IC=0.656)**: Polly is a jompus. Jompuses are impuses. Impuses are amenable. Polly is amenable. True

**Ground-truth Path**: Polly is a jompus. Jompuses are tumpuses. Polly is a tumpus. Tumpuses are yumpuses. Polly is a yumpus. Every yumpus is not amenable. Polly is not amenable. False

---

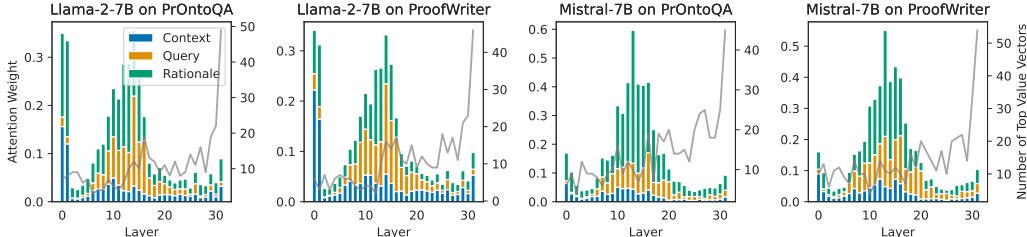

Figure 5: The emergence of internal inconsistency in CoT reasoning could be attributed to **the misalignment between layers with high attention weights on critical tokens and those promotes certain predictions.** The histograms display attention weights for each part (context, query, and rationale) across self-attention layers , accompanied by a gray line indicating the count of value vectors in FFN layers that achieve high cosine similarity to the model's final prediction.

in the last FFN matrix of the layer. Our analysis focuses on the vectors with the top 0.1% of cosine similarity values, which play a pivotal role in forming final predictions. For more details, please see Appendix B.6.

**Results**   Figure 5 presents our findings: 1) Self-attention layers in the middle layers show a marked focus on the query and reasoning steps, a pattern consistent across both models and datasets; 2) FFN layers in the later layers dominate the final model outputs, as indicated by the fact that value vectors with the highest cosine similarities to the probe vector tend to cluster in the later layers. The layers with strong attention on query and reasoning steps and those where high cosine similarity value vectors cluster do not align, providing insight into the possible mechanism behind the emergence of internal inconsistency. This finding is consistent with Figure 2 and Figure 6, suggesting that the model may make correct predictions in the middle layers but fails to fully utilize them in the later layers. To further understand the specific concepts these high-similarity value vectors represent and verify their importance in determining the final prediction, we project them onto the vocabulary space (nostalgebraist, 2020). Additional analysis are provided in Appendix B.6.3.

# 5   Related Work

**Understanding the Inner Workings of Language Models**   The rapid progress in LLM development has necessitated simultaneous efforts to interpret the inner workings of advanced models (Bricken et al., 2023; Ferrando et al., 2024). These works aim to provide an internal view of mechanisms to ensure safety and fairness (Burns et al., 2023; Zou et al., 2023; Li et al., 2024) and to further improve model inference (Schuster et al., 2022; Raposo et al., 2024). Some studies also examine CoT reasoning from an internal perspective and propose methods to teach models to perform implicit CoT (Deng et al., 2023). Unlike these approaches, which require additional training for interpretation, our internal consistency measure offers an off-the-shelf solution to calibrate CoT reasoning, providing a practical and efficient tool for enhancing model reliability.

**Calibration in Language Models**   Traditional calibration methods (Platt et al., 1999; Naeini et al., 2015; Guo et al., 2017) train a parameterized classifier on validation data to adjust the final output of a neural network towards expected outcomes. In the context of LLMs, previous works apply trained parameterized models to the logits at the final layer (Zhao et al., 2021; Shen et al., 2024). In comparison, our study focuses on the phenomenon of internal inconsistency in CoT reasoning and demonstrates that internal consistency is a reliable unsupervised calibration measure.

**Faithfulness of CoT Reasoning**   Although the reasoning capabilities of LLMs have been greatly enhanced with techniques like CoT reasoning (Wei et al., 2022; Yao et al., 2022; Xia et al., 2024), previous investigations into the faithfulness of CoT reasoning have shown that the model's generated rationales often are not consistent with their predictions (Lyu et al., 2023; Lanham et al., 2023). These studies primarily focus on the alignment between a model's verbalized reasoning and its outcomes,

without delving into the model's internal reasoning processes. In contrast, our work provides new insights into how unfaithfulness emerges internally during CoT reasoning and proposes solutions to calibrate reasoning with internal consistency.

Closest to our work is that of Halawi et al. (2023), which studies harmful imitation behaviors of LLMs and the underlying internal mechanisms. Unlike their study that focuses on few-shot learning with false demonstrations, we provide new insights on the phenomenon that CoT reasoning leads to internal inconsistency and that we can calibrate reasoning with internal consistency.

## 6 Conclusions

This work explores the use of internal representations to enhance the reliability of reasoning in LLMs. By examining CoT reasoning, we identify that although generated rationales can improve answer accuracy, they also lead to inconsistencies between middle and final layer representations, potentially affecting reliability. To mitigate this, we introduce internal consistency as a confidence measure, which evaluates the alignment of latent predictions from intermediate layers. Our extensive empirical studies across multiple models and datasets demonstrate that internal consistency is a robust indicator of correct reasoning paths, which we show can be further used for enhancing CoT reasoning by prioritizing paths with high internal consistency. Finally, our analysis on the patterns in attention and feed-forward networks across layers provides insights on why internal inconsistency emerges.

This work has several limitations. First, we restrict our empirical study on decoder-only models. While techniques like decoder lens (Langedijk et al., 2023) is a promising way to extend the concept of internal consistency to encoder-decoder models, we leave it for future work. Second, our analysis focuses on vanilla CoT prompting as the simplest approach for reasoning, whereas many other prompting techniques have been proposed (Gao et al., 2023; Yao et al., 2024). It would be interesting to further investigate internal consistency under these more complex scenarios.

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

## A    Broader Impacts

This work contributes to the understanding and enhancement of reasoning capabilities in LLMs through the lens of internal consistency. While our findings and methodologies offer significant advancements in the calibration and reliability of LLMs, there are potential negative societal impacts to consider. Enhanced reasoning abilities in LLMs could be misused to generate more convincing disinformation, exacerbating issues related to misinformation and manipulation. To mitigate these risks, it is crucial to implement robust monitoring and access control mechanisms. This includes gated release of models, ensuring that only authorized and ethical usage is permitted. Furthermore, developing and providing defenses against misuse, such as mechanisms for detecting and countering disinformation, is essential. By proactively addressing these potential negative impacts, we aim to responsibly advance the field while safeguarding societal interests.

## B    Implementation Details

We performed all experiments on a compute node with 8 Nvidia GPU cards and 512 GB of memory. To replicate our results, the total estimated computing time is under 100 GPU hours, even though the entire research project required more computing power than the experiments detailed in the paper.

### B.1    Preliminary Analysis

We conduct linear probing on activations to investigate how information is processed throughout layers and reasoning steps during CoT reasoning. Specifically, for each reasoning step and intermediate layer, we train a logistic regression classifier on the corresponding hidden states of all data instances. We split the dataset randomly by 80%/20% into training and validation subsets. Following Radford et al. (2021), we use the Scikit-learn package (Pedregosa et al., 2011) and determine the $L_2$ regularization strength $\lambda$ using a hyperparameter sweep over the range between $10^{-6}$ and $10^6$ for logistic regression.

### B.2    Main Experiments

**Models**    Our implementation is based on the Huggingface's Transformer library (Wolf et al., 2019). We use `meta-llama/Llama-2-7b-hf` and `meta-llama/Llama-2-13b-hf` for the Llama series. For the Mistral series, we evaluate `mistralai/Mistral-7B-v0.1` and `mistralai/Mixtral-8x7B-v0.1`. The Llama and Mistral series models are released under the custom license and apache-2.0 license, respectively.

**Dataset Construction**    The construction processes for the test datasets are as follows:

1. BoolQ: We use the original validation set[3], which has 2033 positive and 1237 negative samples. We balance the classes by randomly sampling 1237 samples from the positive ones, resulting in a subset of 2474 samples. BoolQ is released under the Creative Commons Share-Alike 3.0 license.

2. CoinFlip: Since the original dataset used in Wei et al. (2022) is not publicly available, we use the reproduced one[4] created by Kojima et al. (2022) which contains 500 instances.

3. PrOntoQA: We use the official script[5] provided by Saparov and He (2023) to generate 500 instances. Each instance requires 3 hops of reasoning and is based on fictional concept names. PrOntoQA is released under the Apache-2.0 license.

4. ProofWriter: We use the subset[6] provided by Poesia et al. (2023) and randomly sample 500 instances. ProofWriter is released under the CC BY license.

---

[3] https://github.com/google-research-datasets/boolean-questions
[4] https://github.com/kojima-takeshi188/zero_shot_cot/blob/main/dataset/coin_flip/coin_flip.json
[5] https://github.com/asaparov/prontoqa/blob/main/run_experiment.py
[6] https://github.com/gpoesia/certified-reasoning/blob/main/learning/proofwriter/proofwriter_3hop.json

**Prompt Format**  We use the prompt format `{context}\nQ: {query}\nA:` for all datasets and models. For few-shot prompting, in-context examples are prepended using the following format (taking 2-shot prompting as an example): `{icl example 1}\n\n{icl example 2}\n\n{test example}`. For few-shot CoT prompting, we append the rationale after `A:` for each in-context example. Table 3 provides a list of examples for the considered datasets.

**Few-shot Examplars**  We use the few-shot examples from Wang et al. (2022) for BoolQ and from Wei et al. (2022) for CoinFlip. For ProofWriter and PrOntoQA, we directly utilize the examplars from the original datasets.

Table 3: Data examples.

| Dataset | Context | Query |
|---|---|---|
| BoolQ | The myotonic goat, otherwise known as the fainting goat, is a domestic goat that temporarily seizes when it feels panic. If startled by sudden movements or loud noises, they will attempt to escape from the disturbance, generally followed by a startle reaction. In more severe cases, this reaction results in strong tetanic contractions of the agonist and antagonist muscles, causing an uncontrolled stiffness that may cause the goat to remain "frozen" in the position that it was in previous to the attack, or cause it to fall to the ground on its side. During an attack, which may last from 5-20 seconds, the goat can often be picked up without any bending or movement occurring in its body. In the case of goats that are less severely affected with the condition, there may be some minor localized stiffness observed in the legs, however, they are still capable of running away. This behaviour is caused by a hereditary genetic disorder called myotonia congenita. The myotonia goat, similar to humans with congenital myotonia, exhibits no obvious muscle wasting, is rarely incapacitated by the condition, and lives a normal and healthy life span. | True or false: is there such a thing as a fainting goat |
| CoinFlip | A coin is heads up. Whitney flips the coin. Erika does not flip the coin. Tj does not flip the coin. Benito flips the coin. | True or false: Is the coin still heads up? |
| PrOntoQA | Every tumpus is sour. Tumpuses are rompuses. Every rompus is small. Every rompus is an impus. Impuses are floral. Impuses are dumpuses. Dumpuses are not kind. Every dumpus is a yumpus. Each yumpus is feisty. Yumpuses are numpuses. Numpuses are not opaque. Each numpus is a zumpus. Every wumpus is opaque. Every zumpus is temperate. Each zumpus is a jompus. Every jompus is dull. Jompuses are vumpuses. Alex is a dumpus. | True or false: Alex is not opaque. |
| ProofWriter | Fiona is blue. Harry is cold. Harry is white. If Harry is blue and Harry is green then Harry is not round. If something is green then it is young. All white things are young. If something is green and not white then it is not blue. Young, round things are not furry. If something is white and young then it is round. If something is young and not cold then it is round. If something is green and not young then it is not round. | True or false: Harry is round. |

## B.3 Least-to-Most Prompting

In addition to chain-of-thought prompting, we evaluated our approach using least-to-most prompting Zhou et al. (2022) across four datasets. Due to the absence of existing least-to-most prompting templates for these datasets, we developed a systematic process to generate and validate appropriate prompts.

For each dataset, we selected 4 representative instances as our prompt engineering samples. We utilized GPT-4o to generate initial least-to-most decompositions for these instances, followed by manual inspection and refinement. Table 4 presents examples from each dataset demonstrating our least-to-most decomposition approach.

Table 4: Examples of least-to-most prompting decomposition across different datasets.

| Dataset | Context | Decomposition Steps |
|---------|---------|---------------------|
| BoolQ | System of a Down, sometimes shortened to System and abbreviated as SOAD, is an Armenian-American heavy metal band from Glendale, California, formed in 1994. The band currently consists of Serj Tankian (lead vocals, keyboards), Daron Malakian (vocals, guitar), Shavo Odadjian (bass, backing vocals) and John Dolmayan (drums). | 1. Who are the members of System of a Down? 2. What are the roles of each member? 3. How many members are listed as vocalists? |
| CoinFlip | A coin is heads up. Ka flips the coin. Sherrie flips the coin. | 1. What is the starting position of the coin? 2. What happens to the coin after each flip? |
| PrOntoQA | Every tumpus is sour. Tumpuses are rompuses. Every rompus is small. Every rompus is an impus. Impuses are floral. Impuses are dumpuses. Dumpuses are not kind. Every dumpus is a yumpus. Each yumpus is feisty. Yumpuses are numpuses. Numpuses are not opaque. Each numpus is a zumpus. Every wumpus is opaque. Every zumpus is temperate. Each zumpus is a jompus. Every jompus is dull. Jompuses are vumpuses. Alex is a dumpus. | 1. What are the properties of a dumpus? 2. Is Alex a dumpus? 3. Is a dumpus opaque? |
| ProofWriter | Fiona is blue. Harry is cold. Harry is white. If Harry is blue and Harry is green then Harry is not round. If something is green then it is young. All white things are young. If something is green and not white then it is not blue. Young, round things are not furry. If something is white and young then it is round. If something is young and not cold then it is round. If something is green and not young then it is not round. | 1. What are Harry's characteristics? 2. What does being white imply about being young? 3. What does being white and young imply about being round? |

Our decomposition strategy focuses on breaking down complex reasoning tasks into simpler, sequential steps. This systematic approach helps the model handle complex reasoning tasks by addressing simpler sub-problems sequentially.

**Generation** In our few-shot CoT experiments, we use Nucleus sampling (Holtzman et al., 2019) with a temperature of 0.7 and a top-p of 0.95 to generate reasoning paths.

## B.4 The Issue of Miscalibration in Latent Predictions

Ideally, for a balanced dataset with an equal proportion of True and False instances, we would expect the LLM to assign approximately 50% probability to each class. However, our preliminary analysis shows that latent predictions tend to exhibit a skewed distribution, allocating disproportionately high probabilities to one class. Therefore, we implement specific steps to balance the predictions at each layer before use, following Halawi et al. (2023).

Denote the probability assigned by the LLM to the True class as $p(\text{True})$ and the probability assigned to the False class as $p(\text{False})$. We apply the following steps for each layer:

1. We collect the model's output logits for all instances in the dataset.

2. We normalize the probabilities in the $i$-th instance such that $\hat{p}_i(\text{True}) = \frac{p_i(\text{True})}{p_i(\text{True}) + p_i(\text{False})}$.

3. We determine the median of the set $\{\hat{p}_i\}_{i=1}^n$ to be $t$.

4. For any given instance, if the model's output $\hat{p}_i(\text{True}) \geq t$, we classify it as True; otherwise, we classify it as False.

## B.5 Self-attention Layer Analysis

We are interested in the attention value on the context, the query and the generated rationale part. Our calculation can be formally represented as:

$$\text{AttnScore}(\ell, \mathcal{P}) = \frac{1}{H} \sum_{h=1}^{H} \sum_{i \in \mathcal{P}} \text{MHSA}^{(\ell,h)}(i, t_{\text{ans}})$$

where $\text{AttnScore}(\ell, \mathcal{P})$ denotes the average attention weight across the attention heads on layer $\ell$ for part $\mathcal{P}$, $t_{\text{ans}}$ denotes the index of the answer token, $\text{MHSA}^{(\ell,h)}(i, j)$ signifies the sum of attention scores from the $i$-th token to the $j$-th token within the sequence, as computed by the $h$-th attention head at the $\ell$-th layer.

## B.6 FFN Layer Analysis

### B.6.1 Value vector

We utilize the concept of the value vector and the weighted value vector of FFN given by Geva et al. (2022). Specifically, each FFN in layer $\ell$ is made up of two linear transformations with a point-wise activation function in between:

$$\text{FFN}^{\ell}(x^{\ell}) = \sigma(W_K^{\ell} x^{\ell}) W_V^{\ell},$$

where $W_K^{\ell}, W_V^{\ell} \in \mathbb{R}^{d_m \times d}$ are the parameter matrices and $\sigma$ is the activation function. We define the value vector $\mathbf{v}_i^{\ell}$ as the $i$-th column of $W_V^{\ell}$. For an input $x^{\ell}$, the keys produce a vector of coefficients $\mathbf{m}^{\ell} := \sigma(W_K^{\ell} x^{\ell}) \in \mathbb{R}^{d_m}$, then the output of FFN is the weighted average of the value vectors with respect to $\mathbf{m}^{\ell}$, that is:

$$\text{FFN}^{\ell}(x^{\ell}) = \sum_{i=1}^{d_m} m_i^{\ell} \mathbf{v}_i^{\ell}.$$

### B.6.2 Implementation Details

We execute the models on each dataset and extract their final output (True or False) and the last hidden state. Subsequently, we apply logistic regression (without intercept term) in the Scikit-learn package (Pedregosa et al., 2011), and determine the optimal $L_2$ regularization strength $\lambda$ using hyperparameter grid search, based on the last hidden state and the final output of the model. The accuracy evaluated by 5-fold cross-evaluation is above 93% consistently (see Table 5 for details). We observe that using ground truth labels as the training target of logistic regression results in similar result with Figure 5 but the training accuracy is poorer as the model may not know the correct answer.

Then we calculate the cosine similarity of the value vectors and the probe vector. As the average cosine similarity across all layers is relatively noisy because many value vectors are not relevant to the task, we plot the count of value vectors of top 0.1% of the cosine similarity values instead.

Table 5: Cross-validation accuracy of the training of the probe vector.

|  | ProofWriter | PrOntoQA |
|---|---|---|
| Mistral-7B | 97.0% | 97.6% |
| Llama-2-7B | 93.6% | 96.3% |

### B.6.3 Semantics Analysis

To uncover the concepts that the value vectors of the top cosine similarity promote, we project them onto the vocabulary space. Specifically, we are interested in the top tokens of the followings:

- The probe vector, which we refer to as $W_{\text{probe}}$.
- The value vectors with top cosine similarity with the probe vector. We refer the $i$-th value vector at layer $\ell$ as $\mathbf{v}_i^{\ell}$ ($i$ and $\ell$ are indexed from 0).

- We stack the value vectors of top 100 of cosine similarity to a $N \times d$ matrix ($N = 100$) and apply SVD, obtaining the top singular value vectors. We refer the singular value vector of the largest singular value as U.

Filtering out the tokens that are Unicode escape sequences (e.g. \u2705), the result is depicted in Table 6- 9. Note that some token ids encode the same word.

Table 6: Top tokens promoted by the probe vector, value vectors and top singular value vector for Mistral-7B on ProofWriter.

| Vector | Top tokens |
|---|---|
| $W_{\text{probe}}$ | true, true, True, True, sort, right, kind, fin |
| $\mathbf{v}^{19}_{8228}$ | true, truth, TRUE, True, true, correct, True, reality |
| $\mathbf{v}^{26}_{13073}$ | ver, ver, Ver, Vern, verd, VER, verification, verify |
| $\mathbf{v}^{31}_{1549}$ | right, Right, right, Right, RIGHT, droit, Rights, rights |
| $\mathbf{v}^{31}_{597}$ | thinking, really, pretty, constantly, calm, misunder, why, how |
| $\mathbf{v}^{22}_{12854}$ | stronger, infl, elev, loaded, augment, aug, strong, rich |
| $\mathbf{v}^{31}_{2391}$ | most, none, most, none, current, newest, now, more |
| $\mathbf{v}^{21}_{1308}$ | positive, posit, praise, affirm, Pos, triumph, approval, appro |
| $\mathbf{v}^{24}_{5604}$ | right, right, Right, Right, RIGHT, correct, correct, rights |
| U | both, both, Both, beiden, neither, sia, tanto, Neither |

Table 7: Top tokens promoted by the probe vector, value vectors and top singular value vector for Mistral-7B on PrOntoQA.

| Vector | Top tokens |
|---|---|
| $W_{\text{probe}}$ | True, True, Trust, TRUE, TRUE, Tru, true, Truth |
| $\mathbf{v}^{26}_{13073}$ | ver, ver, Ver, Vern, verd, VER, verification, verify |
| $\mathbf{v}^{19}_{8228}$ | true, truth, TRUE, True, true, correct, True, reality |
| $\mathbf{v}^{21}_{1308}$ | positive, posit, praise, affirm, Pos, triumph, approval, appro |
| $\mathbf{v}^{18}_{8216}$ | yes, Yes, Yes, did, does, yes, Roh, rog |
| $\mathbf{v}^{18}_{5555}$ | positive, affirm, strengths, yes, confirmed, endors, Sull, inclusion |
| $\mathbf{v}^{29}_{8705}$ | true, True, true, True, TRUE, TRUE, truly, Tru, false |
| $\mathbf{v}^{18}_{6717}$ | persistence, esso, mero, triumph, /******/, tol, lica, akov |
| $\mathbf{v}^{23}_{9740}$ | eligible, legitimate, compatible, permitted, constitutional, ethical, legit, olv |
| U | FALSE, false, False, absent, FALSE, false, absence, lessness |

Table 8: Top tokens promoted by the probe vector, value vectors and top singular value vector for Llama-2-7B on ProofWriter.

| Vector | Top tokens |
|---|---|
| $W_{probe}$ | True, tr, Tru, Tr, True, TR, true, TRUE |
| $\mathbf{v}^{31}_{6785}$ | t, T, tub, tap, tile, tut, tum, tin |
| $\mathbf{v}^{16}_{1949}$ | pra, triumph, bless, vict, Pra, positive, solutions, rejo |
| $\mathbf{v}^{16}_{10757}$ | confirm, confirm, yes, =" ., confir, otto, dia, unfortunately |
| $\mathbf{v}^{31}_{6356}$ | grande, greater, gradle, groupe, Grad, gradu, great, grand |
| $\mathbf{v}^{20}_{2369}$ | sam, uniform, identical, uniform, hom, similarity, similar, smooth |
| $\mathbf{v}^{16}_{9429}$ | pra, positive, Pos, Pra, posit, happy, Pos, bless, eu |
| $\mathbf{v}^{18}_{722}$ | blo, alive, forced, oku, replacing, eu, active, once |
| $\mathbf{v}^{29}_{3660}$ | ways, ways, MDb, torn, modo, hm, zik, Tourn, etch |
| U | NOT, nicht, W, notin, False, ikke, false, FALSE, moins |

Table 9: Top tokens promoted by the probe vector, value vectors and top singular value vector for Llama-2-7B on PrOntoQA.

| Vector | Top tokens |
|---|---|
| $W_{probe}$ | True, TRUE, TRUE, TR, True, true, TR, truth |
| $\mathbf{v}^{16}_{1949}$ | pra, triumph, bless, vict, Pra, positive, solutions, rejo |
| $\mathbf{v}^{18}_{722}$ | blo, alive, forced, oku, replacing, eu, active, once |
| $\mathbf{v}^{31}_{6356}$ | grande, greater, gradle, groupe, Grad, gradu, great, grand |
| $\mathbf{v}^{16}_{4462}$ | active, straight, bold, walt, visible, Active, Active, dx |
| $\mathbf{v}^{16}_{10757}$ | confirm, confirm, yes, =" ., confir, otto, dia, unfortunately |
| $\mathbf{v}^{16}_{9429}$ | pra, positive, Pos, Pra, posit, happy, Pos, bless, eu |
| $\mathbf{v}^{14}_{8600}$ | pra, positive, esc, salv, escaped, triumph, rag, gem |
| $\mathbf{v}^{25}_{3927}$ | Tr, Tr, tr, tram, trig, tr, trim, trim |
| U | false, FALSE, NOT, FALSE, False, notin, negative, fails |

The result is to verify that the value vectors with the highest cosine similarity to the probe vector indeed encode crucial information for determining the final prediction, so as to verify the emergence of internal inconsistency from the perspective of pattern of attention and FFN layers demonstrated in Section 4.5. This is evidenced by their representation of important keywords such as "true", "false", "verification", and others.

In addition, we also have some interesting discoveries: 1) Some value vectors appear to be of the top cosine similarity with the probe vector in both datasets, indicating their importance and relevance. 2) The value vectors consistently bias towards "True", and the singular value vector of the largest singular value tends to lean towards negative tokens, indicating the necessity of calibration. 3) Some value vectors encode tokens of opposite concepts, which reveals a degree of internal inconsistency within a single value vector.

## C   Additional Results

In this section, we provide additional results to support our analysis. Figure 6 illustrates the patterns of internal consistency across different layers, as discussed in Section 4.2. This figure shows variable convergence patterns across different models and datasets, with significant increases in consistency observed in the final layers. Figures 7-10 present detailed results for each dataset and model corresponding to the findings in Figure 3.

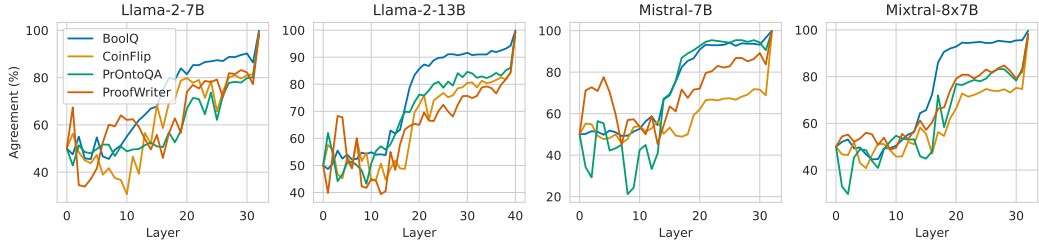

Figure 6: **The patterns of internal consistency are similar across different tasks and models.** The "agreement" value represents the ratio of data instances where the latent predictions match the final predictions. Results are obtained from the zero-shot prompting setting.

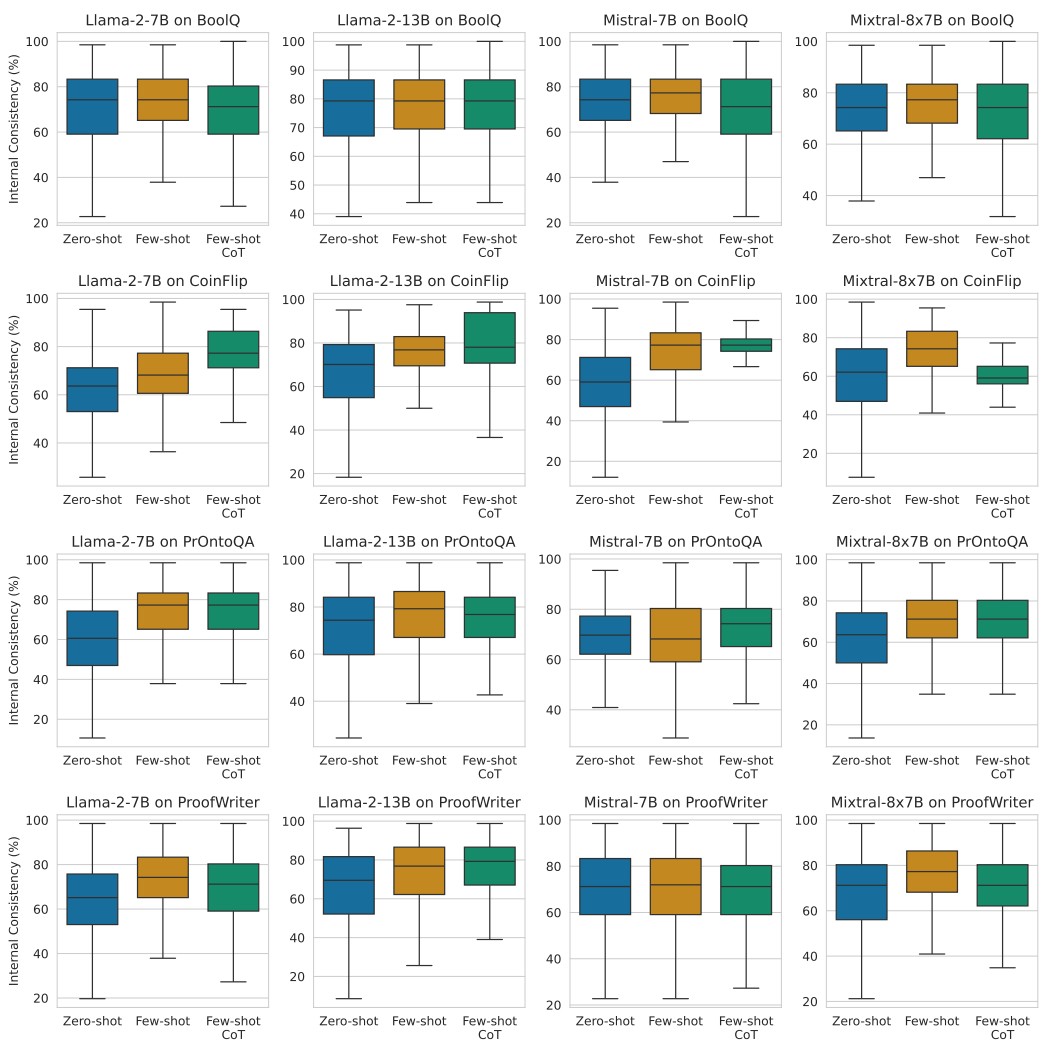

Figure 7: Detailed results for each dataset and model of the first panel in Figure 3: the effect of different prompting techniques on the model's internal consistency.

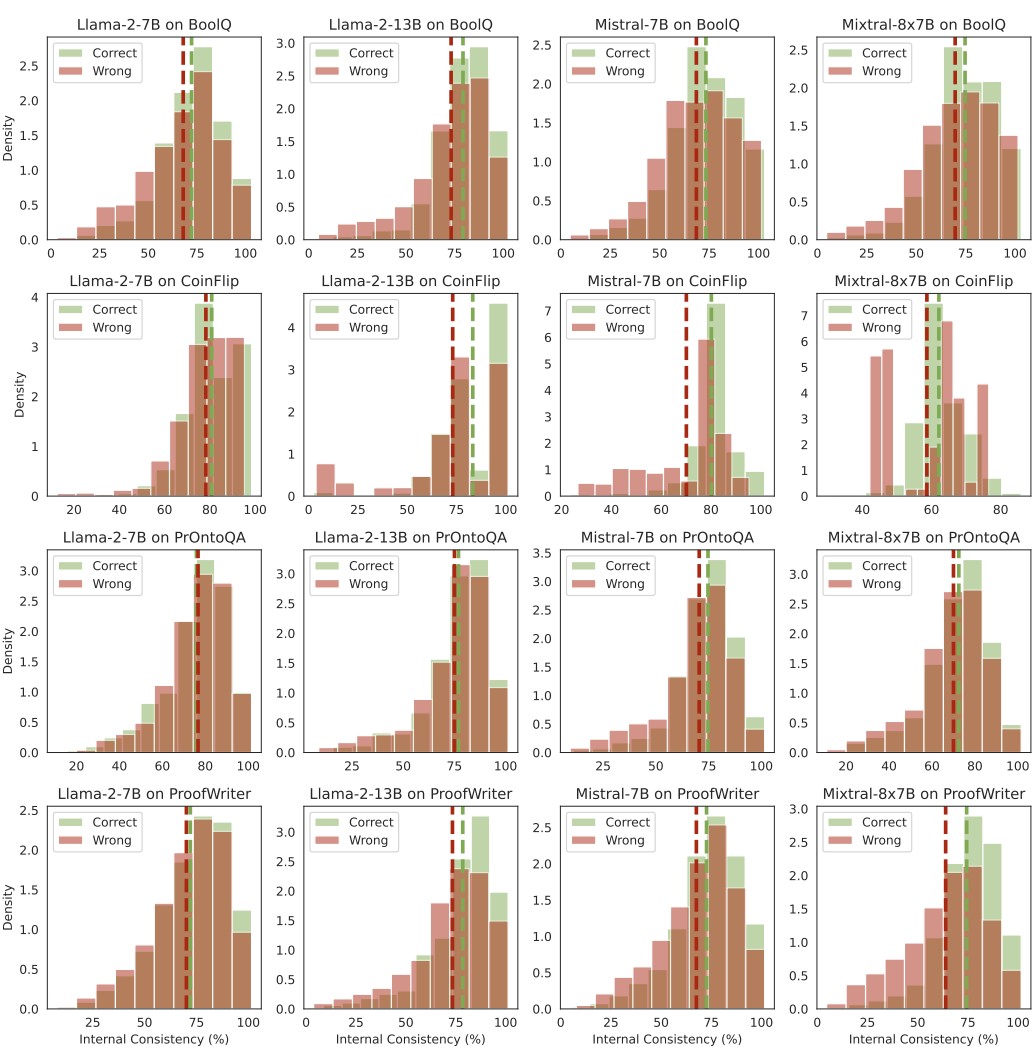

Figure 8: Detailed results for each dataset and model of the second panel in Figure 3: the distribution discrepancy of internal consistency between correct and incorrect model predictions.

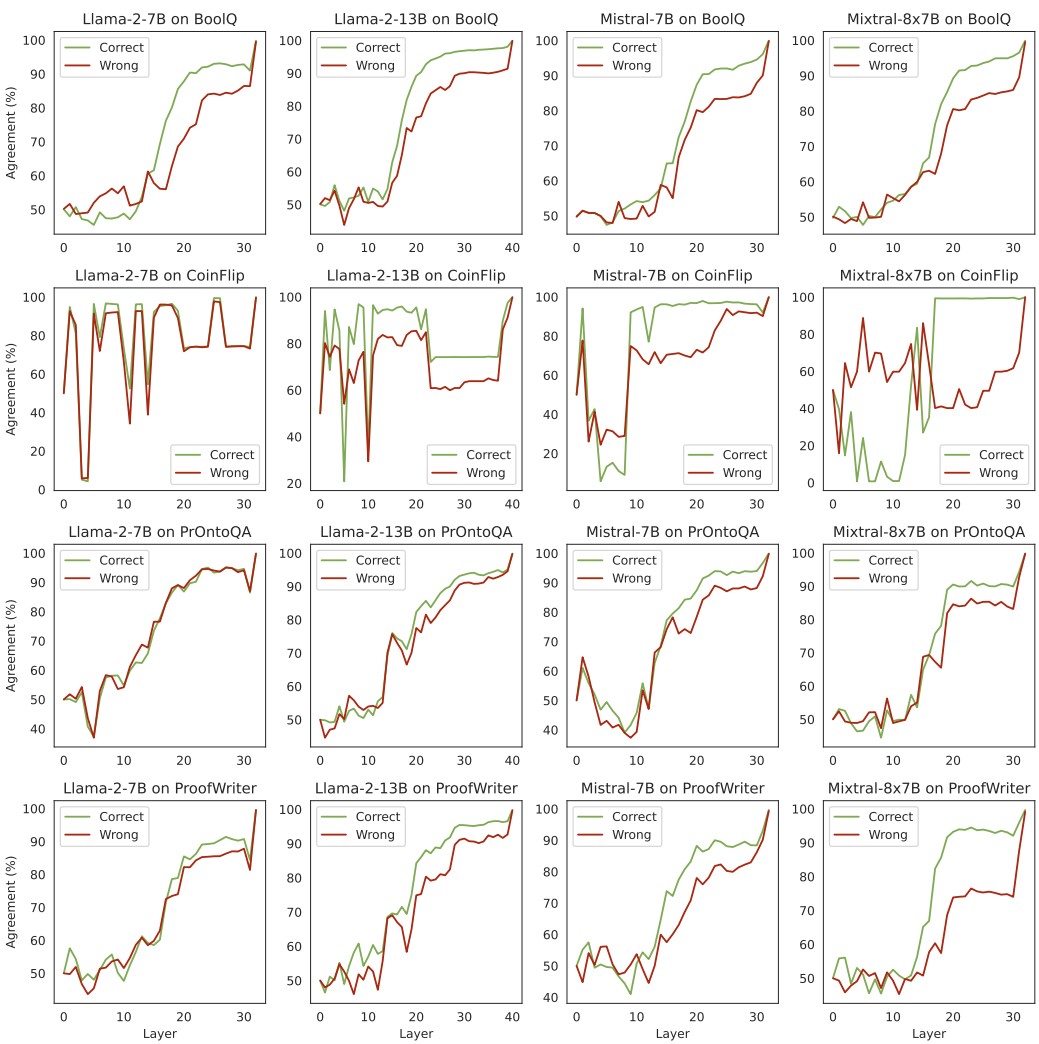

Figure 9: Detailed results for each dataset and model of the third panel in Figure 3: pattern variations in agreement values (representing the ratio of data instances where the latent predictions match the final predictions) across layers.

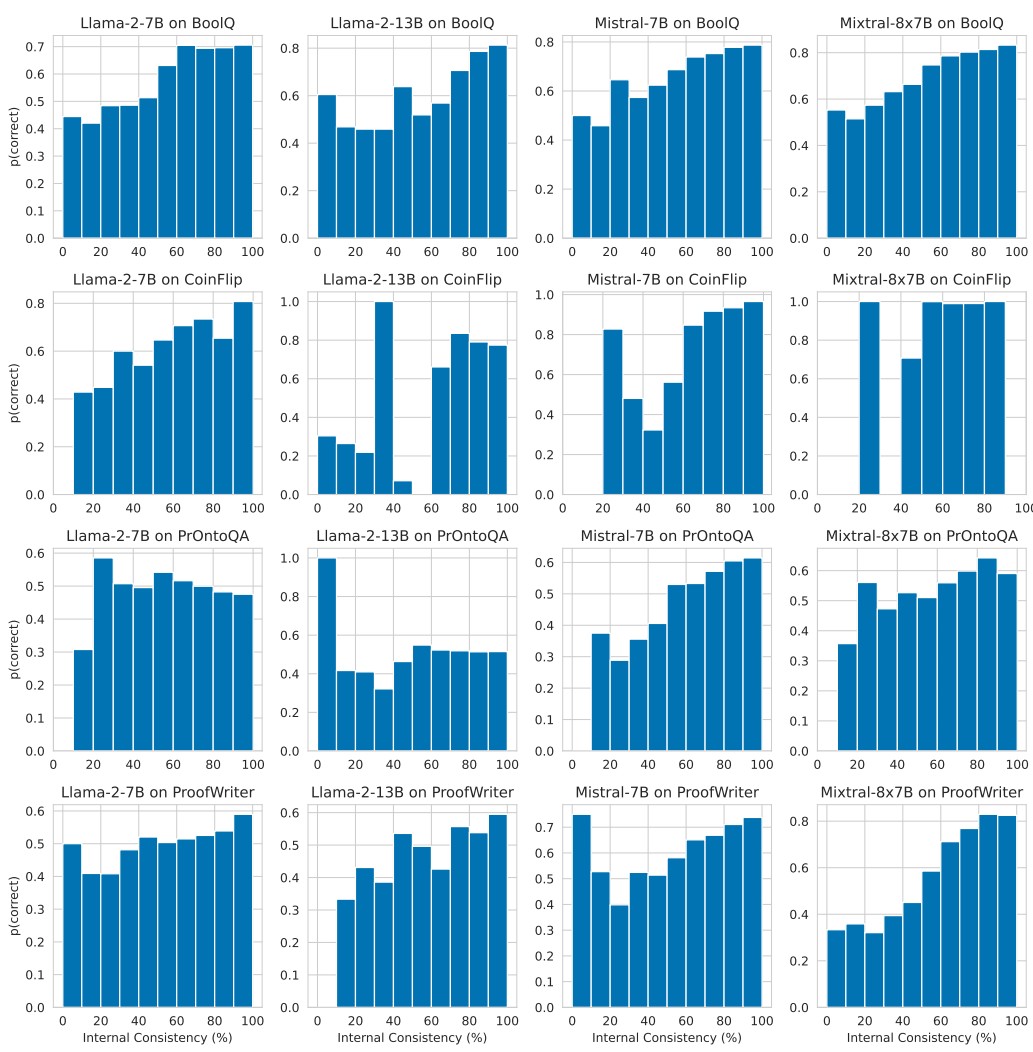

Figure 10: Detailed results for each dataset and model of the fourth panel in Figure 3: a calibration plot with bins according to the model's internal consistency on the x-axis and the accuracy within each bin on the y-axis.

