# OpenReview forum: "Calibrating Reasoning in Language Models with Internal Consistency"
_NeurIPS.cc/2024/Conference — NeurIPS 2024 poster_

### Official Review · Reviewer_fPrR · 2024-07-01

**Soundness:** 4
**Presentation:** 3
**Contribution:** 3
**Rating:** 7
**Confidence:** 4

**Summary:**

This paper investigates the correlation between CoT reasoning and internal representations in Transformer models. It shows that there are inconsistencies between model’s internal representations in different layers. Such inconsistencies can be used to distinguish correct and incorrect reasoning paths, and thereby calibrate model predictions. By incorporating internal consistency into self-consistency (SC+IC), the authors show that SC+IC improves over SC under the same number of sampled reasoning paths. Besides, the authors also provided insights into how internal inconsistency is caused by Transformer components.

**Strengths:**

- The correlation between internal representations and CoT reasoning is a novel topic and the conclusion of this paper would be useful for a wide range of audience working on CoT prompting.
- This paper has a clear logic and the experiment design is solid. The authors first observe inconsistency in internal representations, then propose the internal consistency metric, and show that the metric is correlated with prediction correctness.
- The authors conduct experiments on 4 different datasets and 2 family of models, showing that SC+IC achieves robust improvement over both SC and SC+$\Delta$ at almost no additional cost.

**Weaknesses:**

- The improvement of SC+IC over SC is relatively marginal (+1.2% on avearge). This might be due to the fact that the authors just simply used internal consistency $\in [0, 1]$ as weights in SC. Maybe the authors can try some other weighting strategies (e.g. exponentiating the internal consistency) and obtain better results.
- Section 4.4 is not clear to me. Could you explain why attention weights and number of top value vectors reveal the source of internal inconsistency? Could you elaborate what the probe vector is and how you obtained it? It is not very clear to me even after reading the appendix.
- It is hard to understand some important details (e.g. Figure 6, self-attention layers and FFN layers) without looking into the Appendix. I would recommend the authors to reorganize them in the main paper.

**Questions:**

- Why do we measure internal consistency against the final prediction? How about measuring the consistency between every pair of layers?
- Can internal consistency be applied to non-binary answers? Will it be still effective in that case?
- Line 210: By the term “decreases”, do you mean internal consistency is lower for few-shot CoT than few-shot direct prompting?
- Line 241-242: focusing on the answer tokens → focusing on what the answer tokens attend to.
- Line 248: second → last FFN matrix in the FNN layer. This would be clearer to the readers.
- Figure 5 caption: Are attention weights and number of top value vectors supposed to align? Why?

**Limitations:**

Yes. The authors has discussed the limitations in the paper.

---

> ### Author Rebuttal · Authors · 2024-08-07
>
> Thank you for rating our paper positively\! Your efforts in reviewing our paper and providing profound advice are sincerely appreciated. We would like to address your remaining concerns in the following response.
>
> **W1: Marginal improvement of `SC+IC` over `SC`**
> Thank you for your constructive feedback. We do not observe significant improvement by exponentiating the internal consistency. Nonetheless, we observe significant improvement by applying tuned weights assigned to intermediate layers (please see our global response for details). As shown in Table 1 of the attached pdf, using a weighted voting of latent predictions can drastically improve performance. Perhaps more interestingly, the tuned weights are transferable across datasets, indicating a universal pattern of internal representations for calibration.
>
> **W2: Section 4.4 is unclear**
> As illustrated in the caption of Figure 5, layers with high attention weights on critical tokens do not align with those promoting certain predictions. Such misalignment can result in discrepancy of behavior at different layers, and leads to internal inconsistency. After applying the logistic regression with respect to the last hidden state and the final output (True/False) of the model (as detailed in Appendix B.5.2), the probe vector is the weight matrix (a vector in this case as the labels are binary) of the logistic regression model.
>
> We apologize for any confusion caused and will polish Section 4.4 following your suggestions in the final version.
>
> **W3: Necessary details for the main paper**
> Thank you for your constructive feedback. We will properly reorganize the details and put some important details forward in the main paper for better reading experience.
> **Q1: Why against the final layer?**
> Thank you for your suggestion. We choose the final layer because the final layer gives the natural prediction of the whole model, with minimal modification of the original mechanism of the model. While measuring the consistency between every pair of layers is reasonable, we do not observe improvement in the cost of time complexity.
>
> **Q2: Applicability of IC to non-binary answers?**
> While our analysis in the original paper focuses on datasets with true-or-false answers, the concept of internal consistency can be natually extended to non-binary answers. We add new experiments on SWAG \[1\] and MathQA \[2\]. Please refer to our global response for details. As shown in Table 1 of the attached pdf, our method consistently outperforms the baselines, especially on the more challenging MathQA dataset. Notably, `SC+IC (transfer)` demonstrates robust transferability even though its weights are optimized for binary questions in PrOntoQA, confirming the effectiveness of internal representations for reasoning.
> **Q3-Q5: Writing requiring clarification**
>
> Line 210 “decrease”: Actually we mean longer reasoning paths tend to result in lower internal consistency, as illustrated in Section 2.2 and Figure 2\.
>
> Line 241-242 and line 248: Thank you for your advice. Your rephrasing is much clearer than our original expression and we will correct them following your advice.
>
> **Q6: Confusion about Figure 5's caption**
>
> Actually we do not expect or suppose the attention weights and number of top value vectors to align. Section 4.4's analysis seeks to explore how transformer components contribute to internal consistency, with the misalignment between these factors offering insight into why internal inconsistency arises.
>
> \[1\] Zellers R, Bisk Y, Schwartz R, et al. Swag: A large-scale adversarial dataset for grounded commonsense inference\[J\]. arXiv preprint arXiv:1808.05326, 2018\.
> \[2\] Amini A, Gabriel S, Lin P, et al. Mathqa: Towards interpretable math word problem solving with operation-based formalisms\[J\]. arXiv preprint arXiv:1905.13319, 2019\.

---

> > ### Comment · Reviewer_fPrR · 2024-08-10
> >
> > Thanks the authors for their response. I am happy to see that with tuned weights, SC+IC can achieve much better performance, and this transfers across datasets. I decide to raise my score from 6 to 7.

---

> > > ### Author Response · Authors · 2024-08-11
> > >
> > > We are pleased that you find our new experiments helpful and appreciate your insightful review. Please feel free to ask any further questions.

---

### Official Review · Reviewer_reW1 · 2024-07-08

**Soundness:** 3
**Presentation:** 2
**Contribution:** 3
**Rating:** 6
**Confidence:** 3

**Summary:**

This paper first reveals that inconsistencies emerge between the internal representations in middle layers and those in final layers, which might undermine the reliability of reasoning processes, and then proposes internal consistency to measure the model’s confidence. Furthermore, the authors propose a method to calibrate CoT reasoning by up-weighting reasoning paths with high internal consistency, which boosts reasoning performance.

**Strengths:**

- This paper identifies the inconsistencies between intermediate and final layer representations in LLM reasoning, and proposes a measure of internal consistency to evaluate the model’s confidence and calibrate CoT reasoning.
- Extensive results across different models and datasets show that the proposed measure can effectively distinguish between correct and incorrect reasoning processes.
- The authors propose a new method to calibrate CoT reasoning by up-weighting reasoning paths with high internal consistency, thereby improving reasoning performance.

**Weaknesses:**

- The motivation for this paper doesn't sound convincing enough (see Questions part).
- The improvements in Table 1 seem to be limited to some models/datasets. Although the authors performed ten runs with different random seeds, they mentioned that each dataset contains at least 500 samples, which means that the results may vary between 0-10 samples. I suggest the authors report detailed data statistics.
- This paper includes extensive experiments and results, but lacks analysis and description; Readers need to observe for themselves, such as Figure 3.
- In Figure 3 (a), the text on the x-axis is overlapping.
- For Figure 2 (b) and (d), it is unclear how the different internal consistency results (0, 25, 50, 75, 100) were obtained. For example, the authors mentioned that results are averaged over all models and datasets, so (a) shows results corresponding to different prompting methods, and (c) shows results corresponding to different layers.
- The authors describe Figure 6 in the main text, but the figure is placed in the appendix (of course this may be improved in the final version).

**Questions:**

- Intuitively, as intermediate layers get closer to the final layer, their predictions should increasingly match the final predictions. Why do you think they need to be internally consistent?

**Limitations:**

see above.

---

> ### Author Rebuttal · Authors · 2024-08-07
>
> Thank you for rating our paper positively\! We sincerely appreciate your observant review and insightful suggestions and aim to address your concerns in the following response.
>
> **W1: Motivation behind internal consistency**
> We would like to clarify that we do not expect latent predictions to be internally consistent, but to use consistency as a measurement for calibration. More specifically, our study on internal consistency is motivated by our preliminary analysis that inconsistency emerges at middle layers and is exacerbated in CoT reasoning. This leads to an intuition that predictions with greater certainty can converge earlier in latent predictions. Consequently, we emphasize consistency from a comparative perspective: A stronger consistency between the latent predictions at intermediate layers and the final prediction indicates higher certainty and thus can be used to weigh reasoning paths for CoT.
>
> We apologize for any confusion caused by insufficient clarity regarding our motivations and will revise these sections for improved clarity in the final version.
>
> **W2: The improvements in Table 1 seem to be limited to some models/datasets & Missing detailed data statistics**
>
> We would like to highlight that, although the improvement is relatively small for datasets such as BoolQ, consistent performance enhancements are evident across various models and datasets. The gains are particularly pronounced in more challenging reasoning tasks, such as logical reasoning. Given that `IC` is an unsupervised method, we consider these results valid in demonstrating the calibration effectiveness of internal consistency.
>
> Our method can achieve further improvements by applying tuned weights assigned to intermediate layers. As shown in Table 1 of the attached pdf, using a weighted voting of latent predictions can drastically improve performance. Perhaps more interestingly, the tuned weights are transferable across datasets, indicating a universal pattern of internal representations for calibration.
>
> Regarding detailed data statistics, we place them in the Appendix B.2 of the original paper, including specific numbers of samples and other details.
>
> **W3: Results lack analysis and description**
> We apologize for any confusion caused by the lack of analysis and are happy to discuss with you on specific contents that you feel unclear.
>
> For Figure 3, the panels sequentially display: a) The boxplot describing the distribution of internal consistency under each prompting setting, highlighting the emergence of inconsistency in CoT setting. b) The distribution of internal consistency level when the model gives correct and wrong predictions, and the discrepancy suggests internal consistency can be an indicator of prediction accuracy. c) The discrepancy of the ratio of data instances where the latent predictions match the final ones across layers, which further illustrates incorrect reasoning paths lead to lower consistency level. d) A plot describing the prediction accuracy under each internal consistency level, shows that internal consistency is a reliable metric for calibration. We will elaborate the description in the final version.
> **W4-W6: Issues with the figures**
> Thank you for your constructive feedback\! We will reorganize the figures in our final version based on your suggestions.
>
> Regarding Figures 3(b) and 3(d), we calculate internal consistency for each sampled reasoning path and associate these values to the accuracy of the final predictions. The results are presented as histograms, where each bar corresponds to a specific bin.

---

> > ### Comment · Reviewer_reW1 · 2024-08-12
> > **Reviewer Response to Authors' Rebuttal**
> >
> > Dear authors,
> >
> > Thanks for your response! I decided to maintain my score.

---

> > > ### Author Response · Authors · 2024-08-13
> > >
> > > We greatly appreciate your feedback. Please feel free to ask any further questions!

---

### Official Review · Reviewer_7a9f · 2024-07-12

**Soundness:** 2
**Presentation:** 3
**Contribution:** 2
**Rating:** 5
**Confidence:** 4

**Summary:**

This paper investigates the use of internal consistency as a measure for calibration, and to improve the reasoning capabilities of LLMs. The internal consistency is a measure of the model’s confidence by examining the agreement of latent predictions decoded from intermediate layers, and this measure does not require additional training. The authors conduct extensive empirical studies across various models and datasets, demonstrating that internal consistency effectively distinguishes between correct and incorrect reasoning paths.

**Strengths:**

1. The proposed measurement is practical, offering a new way to self-evaluate and calibrate reasoning without additional training or human annotations.
2. The authors conduct extensive empirical studies across different models and datasets.
3. This paper is also well-structured and clearly presents its methodology, experiments, and findings.

**Weaknesses:**

1. The proposed measurement is rather straightforward, using an indicator function to measure if latent predictions match the final prediction and summing over all intermediate layers.
2. The authors claim that the proposed measure is more accurate than logit-based methods and does not require additional training. However, Table 1 does not show a significant performance gap.
3. The evaluation is conducted on specific datasets involving true-or-false questions, which may not fully represent the diversity of reasoning tasks encountered in practical applications.
4. The paper dedicates considerable space to explaining the Transformer architecture, which seems unnecessary given the target reader's background. The listed equations for Transformer architecture and internal representations are not utilized in later sections. In contrast, the authors should elaborate more on the decoding process using the proposed consistency score, which is only briefly mentioned.

**Questions:**

1. Will the authors provide more probing results regarding the suitability of lower layers for prediction and the proposed measurement?
2. Will the authors include results using other approaches for reasoning?

**Limitations:**

Yes

---

> ### Author Rebuttal · Authors · 2024-08-07
>
> Thank you for the comprehensive and insightful feedback\! We appreciate your effort in reviewing our paper and are committed to addressing your concerns in the following response.
>
> **W1: Internal consistency is rather straightforward**
> Indeed, internal consistency (IC) serves as a straightforward measurement for calibration. We consider simplicity a valuable aspect of our approach, given that it incurs minimal computational overhead (approximately 0.01% additional time as discussed in the response to Reviewer B2Yn) and can be seamlessly integrated into various models without the need for tuning.
>
> To better utilize internal consistency for calibration measurement, we conduct new experiments that tune the weights assigned to intermediate layers during IC calculation. For a comprehensive setup and analysis of the results, please see our global response and our response to W2.
>
> **W2: Table 1 does not achieve great improvement**
> We note that `SC+IC` is an unsupervised method that is not tuned on specific data. While the improvements are not substantial, we observe consistent performance gains over `SC` across models and datasets, especially on tasks that require more complex reasoning.
>
> The improvements can be lifted by applying tuned weights assigned to intermediate layers. As shown in Table 1 of the attached pdf, using a weighted voting of latent predictions can drastically improve performance. Perhaps more interestingly, the tuned weights are transferable across datasets, indicating a universal pattern of internal representations for calibration.
>
> **W3: True-or-false questions may not fully represent the diversity of reasoning tasks**
> Following a similar analytical protocol in previous work \[1, 2\], we focus on the true-or-false questions for evaluation in the original paper. To increase the diversity of reasoning tasks, we add new experiments on SWAG \[3\] and MathQA \[4\]. As shown in Table 1 of the attached pdf, our method consistently outperforms the baselines, especially on the more challenging MathQA dataset. Notably, `SC+IC (transfer)` demonstrates robust transferability even though its weights are optimized for binary questions in PrOntoQA, confirming the effectiveness of internal representations for reasoning.
>
>
> **W4: Paper organization**
> Thank you for your constructive feedback\!
>
> The background section on the Transformer architecture aims to establish notations and introduce readers who are unfamiliar with it. However, we acknowledge that this section is overly lengthy and contains redundant information. In the final version, we will reorganize the structure to enhance clarity and presentation.
>
> **Q1: More probing results regarding the suitability of lower layers**
> We apply probing in our preliminary analysis of internal consistency. While the pattern observed is valid, we find that probing cannot reliably indicate subtle differences in lower layers–a limitation inherent to probing \[5\]. Therefore, we instead focus on CoT experiments for evaluation.
>
> Regarding the suitability of lower layers, please refer to our detailed response to Q1 from Reviewer B2Yn, where we present new experimental results and discuss the use of lower layers.
>
> **Q2: Results using other approaches for reasoning**
> We conduct new experiments using least-to-most (L2M) prompting \[6\], a representative method other than chain-of-thought reasoning to elicit reasoning. We sample 4 instances from each dataset (8 for coinflip) as in-context learning examples, following the setup in our original paper, and use GPT-4o to generate subproblems and answer to each subproblem. We manually check the prompts to ensure they are correct. The results below indicate that integrating IC into L2M prompting consistently improves reasoning performance, confirming the efficacy of our approach.
>
> |  | L2M | SC | SC+$\\Delta$ | SC+IC | SC+IC (tune) |
> | :---- | :---- | :---- | :---- | :---- | :---- |
> | Llama2-7B | 63.8 | 69.8 | 70.3 | 71.0 | 73.4 |
> | Llama2-13B | 73.5 | 79.1 | 79.6 | 79.4 | 80.8 |
> | Mistral-7B | 74.6 | 78.4 | 79.2 | 79.1 | 81.2 |
> | Mixtral-8x7B | 78.7 | 83.1 | 83.6 | 83.2 | 84.4 |
>
> *Results of least-to-most prompting. We report calibrated accuracy averaged over four datasets.*
>
> \[1\] Li K, Patel O, Viégas F, et al. Inference-time intervention: Eliciting truthful answers from a language model\[J\]. Advances in Neural Information Processing Systems, 2024, 36\.
> \[2\] Halawi D, Denain J S, Steinhardt J. Overthinking the truth: Understanding how language models process false demonstrations\[J\]. arXiv preprint arXiv:2307.09476, 2023\.
> \[3\] Zellers R, Bisk Y, Schwartz R, et al. Swag: A large-scale adversarial dataset for grounded commonsense inference\[J\]. arXiv preprint arXiv:1808.05326, 2018\.
> \[4\] Amini A, Gabriel S, Lin P, et al. Mathqa: Towards interpretable math word problem solving with operation-based formalisms\[J\]. arXiv preprint arXiv:1905.13319, 2019\.
> \[5\] Alain G, Bengio Y. Understanding intermediate layers using linear classifier probes\[J\]. arXiv preprint arXiv:1610.01644, 2016\.
> \[6\] Zhou D, Schärli N, Hou L, et al. Least-to-most prompting enables complex reasoning in large language models\[J\]. arXiv preprint arXiv:2205.10625, 2022\.

---

> > ### Comment · Reviewer_7a9f · 2024-08-10
> > **Reply to authors' rebuttal**
> >
> > Thanks for the authors' response. WIth additional experiments and generalized results demonstrated, I decide to raise my score.

---

> > > ### Author Response · Authors · 2024-08-11
> > >
> > > We greatly appreciate your recognition of our rebuttal and your thorough review. Feel free to raise any follow-up questions!

---

### Official Review · Reviewer_B2Yn · 2024-07-13

**Soundness:** 3
**Presentation:** 3
**Contribution:** 3
**Rating:** 6
**Confidence:** 3

**Summary:**

The paper explores the concept of "internal consistency" in Large Language Models (LLMs), mainly focusing on chain-of-thought (CoT) reasoning, where models articulate step-by-step rationales. The authors probe internal representations of LLMs to assess the alignment between middle and final layer outputs and propose boosting reasoning paths with higher internal consistency. Through studies across various models and datasets, authors demonstrate that models enhanced with internal consistency metrics reach significant improvement on performances in reasoning tasks.

**Strengths:**

1. Leveraging internal consistency within LLMs to improve reasoning performance is novel and promising. It addresses the issue of LLMs producing reasoning paths that are not always aligned with their final predictions.
2. The paper provides validations across multiple models and datasets, enhancing the confidence of the results.
3. By demonstrating how internal consistency can be used to calibrate CoT reasoning, the paper contributes practical techniques for improving LLM reliability, which is crucial for applications requiring precise and trustworthy outputs.

**Weaknesses:**

1. The techniques described require accessing intermediate layer outputs and could be computationally expensive, limiting the scalability of the proposed methods, especially in resource-constrained environments.
2. There is a risk that optimizing for internal consistency might lead models to overfit specific reasoning patterns or datasets, potentially reducing the generalizability of the findings.

**Questions:**

1. Can you briefly explain why low LLM layers (<10) can be used to show internal consistency?
2. With high internal consistency, the results from high LLM layers (>20) may be the same as the last layer. Can we remove the last few layers to reduce the computational complexity of LLMs and increase their reasoning speed? In other words, can we use the method to reduce the size of LLMs?
3. How does internal consistency interact with other known issues in LLMs, such as hallucination or bias in data?

**Limitations:**

This paper appropriately describes its limitations.

---

> ### Author Rebuttal · Authors · 2024-08-07
>
> Thank you for your thoughtful reviews\! We appreciate your recognition of our method as “novel and promising”. We aim to address your concerns in the following response.
>
> **W1: The proposed techniques could be computationally expensive**
> The proposed techniques incur minimal computational overhead compared to decoding. To assess internal consistency (IC), we apply the model's unembedding matrix to the hidden states of each intermediate layer associated with the answer token, once per reasoning path. Given that these hidden states are computed during decoding, they can be repurposed for evaluating IC without additional computation.
>
> For a more detailed analysis, we estimate the proportion of computation dedicated to assessing IC in Llama-2-7b. The FLOPS per token during decoding amount to approximately $2P$, where $P$ represents the total number of parameters. For calculating IC, this is bounded by the normalization step prior to unembedding, with a time complexity of $O(dL)$, with $d$ being the hidden size and $L$ being the number of layers. Since $dL \\ll P$, the computation overhead is negligible. It is important to note that because IC is computed only once per reasoning path, the actual proportion of allocated computation is very low (\~0.01% of the time required to decode a reasoning path). Below are statistics from one of our setups:
>
> |  | IC Calculation | Decoding per Token | Decoding per Path |
> | :---- | :---- | :---- | :---- |
> | Time (ms) | 0.249648 | 34.9162 | 2283.64 |
>
> *Time complexity comparison between IC calculation and decoding; benchmarked with Llama-2-7B on PrOntoQA.*
>
> **W2: The overfitting risk**
> We would like to clarify that our method requires no training process to calculate IC and apply IC in reasoning. Without tuning on specific data, our method consistently achieves better results over baselines on various models and datasets, validating the generalizability of our method.
>
> In response to your interest in potential overfitting risks when optimizing for specific data, we conduct additional experiments on tuning weights assigned to each layer during IC calculation (see a more detailed setup in our global response). Despite the minimal number of parameters ($L$, the number of layers) involved, tuning weights yields substantial enhancements on various datasets and models. Importantly, the optimized weights are also adaptable to different datasets, indicating a universal pattern in internal representations that can be leveraged across multiple tasks.
>
> **Q1: Why use lower layers?**
> The use of lower layers is motivated by the intuition that predictions with greater certainty can converge earlier in latent predictions. This intuition is supported by Figure 3c, which shows that lower layers exhibit a higher agreement rate for correct predictions than for wrong ones. Consequently, we emphasize consistency from a *comparative perspective*: A stronger consistency between the latent predictions of lower layers and the final prediction indicates higher certainty and thus can be used to weigh reasoning paths.
>
> Indeed, Figure 3c also suggests that lower layers (\<10) contain less semantic information and are relatively noisy. We conduct a new experiment to exclude lower layers for IC calculation, which results in an improvement for Llama-2 models. However, we also note that it introduces the need to specify layers to exclude and we plan to discuss it along with the new tuning results in the revised version.
>
> |  | Llama-2-7B | Llama-2-13B | Mistral-7B | Mixtral-8×7B |
> | :---- | :---- | :---- | :---- | :---- |
> | SC+IC | 62.4 | 70.6 | 76.1 | 80.6 |
> | SC+IC (w/o lower) | 62.8 | 71.1 | 76.2 | 80.7 |
>
> *Comparison of calculating IC on all layers versus on layers higher than 10\. We report calibrated accuracy averaged over different datasets.*
>
> **Q2: Removing the last few layers to reduce computational complexity?**
> Yes, it is possible to reduce computational complexity by implementing an early-exiting strategy based on IC, using a proper stopping criterion like saturation \[1\]. Nonetheless, our primary aim is to explore how IC can improve reasoning capabilities rather than accelerate inference, which has been the focus of prior research \[1, 2\].
>
> **Q3: Interaction between internal consistency with issues like hallucination or bias in data?**
> IC serves as a metric for calibration, aiding LLMs in expressing their confidence in the prediction. This is essential for mitigating issues like hallucination or bias, by detecting possible hallucinated or biased predictions with IC and correcting them accordingly \[3\]. In this regard, our approach to calibration in reasoning aligns well with hallucination and bias detection. We will provide more detailed discussion on this interaction in the final version.
>
> \[1\] Schuster T, Fisch A, Gupta J, et al. Confident adaptive language modeling\[J\]. Advances in Neural Information Processing Systems, 2022, 35: 17456-17472.
> \[2\] Raposo D, Ritter S, Richards B, et al. Mixture-of-Depths: Dynamically allocating compute in transformer-based language models\[J\]. arXiv preprint arXiv:2404.02258, 2024\.
> \[3\] Pan L, Saxon M, Xu W, et al. Automatically correcting large language models: Surveying the landscape of diverse self-correction strategies\[J\]. arXiv preprint arXiv:2308.03188, 2023\.

---

> ### Comment · Reviewer_B2Yn · 2024-08-12
> **Reply to the Rebuttal**
>
> Thanks for the authors' response. Given the additional experiments, generalized results, and answers to my questions, I decided to raise my score.

---

> > ### Author Response · Authors · 2024-08-12
> >
> > We greatly appreciate your thorough review and are glad that our rebuttal was helpful. We are happy to address any further questions you may have.

---

### Author Rebuttal · Authors · 2024-08-07

We greatly appreciate the detailed feedback from the reviewers. Your insightful suggestions have significantly inspired us to enhance our draft. We are committed to address the reviewers’ concerns on a point-by-point basis. Regarding the common concerns raised by the reviewers, we have conducted additional experiments:

**New results on multi-choice SWAG and MathQA datasets**. We sample the first 500 instances from the train split of each dataset and follow the same setup in the original paper to conduct chain-of-thought experiments. These datasets with non-binary answers naturally extend the evaluations and support the effectiveness of our method.
**New results on variants of SC+IC.** We consider a weighted voting method for IC in Section 3.2: Instead of assigning equal weights to different layers, we apply a weight vector $\\mathbf{w} \\in \\mathbb{R}^{L}$ for aggregation. We consider two variants `SC+IC (tune)` and `SC+IC (transfer)`. For each dataset, we use a small number of 500 held-out samples to tune weights assigned to intermediate layers for internal consistency calculation. We use the Adam optimizer and a learning rate of 0.01 with 1000 optimization steps. For `SC+IC (transfer)`, we apply the same weights tuned for PrOntoQA to different datasets. We observe significant improvement and great transferability, albeit that the number of introduced parameters is only the number of layers.

The additional results are presented in the attached pdf. In Table 1, we provide an updated version of our main results including the new datasets and variants. In Figure 1, we show the tuned weights for different models, showing a relatively important role of middle layers in contributing the effectiveness of internal consistency.

---

### Decision · Program_Chairs · 2024-09-25

**Decision:**

Accept (poster)

**Comment:**

The authors evaluate internal representations to align middle and final layer outputs, suggesting improvements to reasoning paths using higher internal consistency. Studies across various models and datasets show that enhancing internal consistency metrics significantly boosts performance in reasoning tasks.
Reviewers agree that the novelty of the internal consistency idea.